# Facile and Green Process to Synthesize a Three-Dimensional Network Few-Layer Graphene/Carbon Nanotube Composite for Electromagnetic Interference Shielding

**DOI:** 10.3390/polym14091892

**Published:** 2022-05-05

**Authors:** Yu-Hong Yeh, Kuei-Ting Hsu, Chia-Hung Huang, Wei-Ren Liu

**Affiliations:** 1Department of Chemical Engineering, R&D Center for Membrane Technology, Chung Yuan Christian University, 200 Chung Pei Road, Chungli District, Taoyuan City 320, Taiwan; ziv910543@gmail.com; 2Department of Chemical Engineering, Army Academy, No.750, Longdong Rd., Chung-Li 320, Taiwan; 3Department of Electrical Engineering, National University of Tainan, No.33, Sec. 2, Shulin St., West Central District, Tainan City 700, Taiwan; chiahung@mail.mirdc.org.tw; 4Metal Industries Research and Development Centre, Kaohsiung 701, Taiwan

**Keywords:** graphene, carbon nanotubes, electromagnetic interference shielding

## Abstract

We propose an environmentally friendly liquid exfoliation approach and subsequent freeze-drying process for constructing a three-dimensional (3D) carbon-based network by using few-layer graphene (FLG) and carbon nanotubes (CNTs) for electromagnetic interference (EMI) shielding applications. Systematic characterizations—such as X-ray diffraction, scanning electron microscopy, and transmission electron microscopy—as well as Raman characterization and EMI shielding tests were performed. The results indicated that the as-synthesized 3D-FLG/CNT composite obtained through the freeze-drying process exhibited excellent electromagnetic interference shielding. The shielding effect of FLG could be improved from 15 to 22 dB by introducing CNTs. The CNTs inhibited restacking of FLG in the structure. We also compared two drying processes: oven drying and freeze-drying. The freeze-drying technique markedly improved the shielding effect of FLG/CNTs from 22 to 36 dB. The composition-optimized 3D-FLG/CNT composite could be a candidate material for use in EMI shielding.

## 1. Introduction

Electronic products have become increasingly widespread in daily life. Nevertheless, electronic devices emit electromagnetic waves that are invisible and harmful to the human body [1,2,3,4,5,6,7,8]. Being in an environment with many such devices for a long period may lead to a concern for health. Thus, the development of electromagnetic-wave-shielding materials to reduce or even eliminate electromagnetic waves is crucial [9,10,11,12]. In earlier days, the widely suggested shielding material was metal [13], which has a strong electromagnetic-wave-shielding effect (SE) due to its high electrical conductivity [14,15,16,17]. However, metal-based materials may undergo corrosion and oxidation. Their higher density also limits their application [18]. The most critical disadvantage is that the shielding offered by metal originates from its reflection efficiency [19,20]. The creation of secondary reflected electromagnetic waves does not solve the problem and can damage electronic devices and shorten their lifetime. Carbon-based materials are promising alternatives to metal due to their lightness, corrosion resistance, flexibility, and high conductivity [21,22,23,24,25,26,27,28]. Carbon-based materials such as few-layer graphene (FLG) and carbon nanotubes (CNTs) have been demonstrated to be potential electromagnetic-wave-shielding materials [29,30,31,32,33,34,35,36,37,38,39,40,41,42,43,44,45,46,47,48,49,50].

Liu et al. used chitosan to improve the microporous structure of the interface between graphene sheets, thereby increasing the conductivity of the modified graphene/iron pentacarbonyl porous membrane. The electromagnetic wave-SE was as high as 38 dB. The SE was 6.3 times higher than that of reduced graphene oxide/epoxy (6 dB). The conductivity of the chitosan-modified material (40.2 S/m) was 1340 times higher than that of the unmodified material (0.03 S/m) [36]. Fu et al. reported a highly flexible single-walled CNT (SWCNT)/graphene film. After folding 1000 times, the SWCNT/graphene film maintained excellent mechanical and electrical stability. A structure for multiple reflections was constructed between layers of SWCNT/graphene films to produce an excellent EMI SE of approximately 80 dB [37]. Bagotia et al. successfully synthesized a synergistic graphene/CNT hybrid nanostructure by connecting multiwalled CNTs (MWCNTs) between graphene layers; with the construction of a polycarbonate/vinyl methyl acrylate graphene MWCNT structure, this hybrid improved mechanical properties, electrical conductivity, and EMI shielding performance. The highest conductivity of this structure was 1.91 × 10^−1^ S/cm. The corresponding electromagnetic SE was 34 dB [38]. Zhang et al. developed foam comprising thermally reduced graphene oxide/CNTs (~0.65 g/cm^3^); this foam had a microporous structure and exhibited excellent conductivity of 2.92 S/m and high EMI SE of 30.4 dB [39]. Zhu et al. used graphene and oxide CNTs (OCNTs) as precursors to construct a 3D conductive framework through simple and convenient filtration and calcination methods. For polydimethylsiloxane-coated r-OCNT/thermally-reduced graphene composite filter cake with a thickness of approximately 1 mm, a strong EMI SE of 67.3 dB was obtained in the frequency range of 8.2–12.4 GHz. After repeated bending 10,000 times, the SE of the composite material was still 62.7 dB [40].

In this paper, we propose a simple and facile jet cavitation process for synthesizing FLG and FLG/carbon nanotube (FLG/CNT) suspensions. One-dimensional CNTs were introduced into the skeleton of FLG to prevent restacking of FLG during drying processes. We compared two drying methods: oven drying and freeze-drying. The results indicated that freeze-dried samples had a highly porous structure and exhibited the highest EMI shielding performance of 40 dB.

## 2. Materials and Methods

### 2.1. Materials and Chemicals

Graphite was purchased from Knano Graphene Technology (Xiamen, China). MWCNTs were purchased from Jiangsu Cnano Technology (Zhenjiang, China). Paraffin wax was purchased from Sigma-Aldrich (Burlington, MA, USA).

### 2.2. Preparation of FLG/CNTs/Wax Composite

Figure 1 displays the schema of the FLG, graphene nanotube (GNT) (FLG/CNT), FFLG (freeze-dried FLG), and FGNT (freeze-dried FLG/CNT) composite materials. The lower-left corner shows the FLG/CNT/wax composite. Graphite and carbon nanotubes were added to deionized water to attain a solid content of 5 wt.%. The liquid exfoliation process of graphite was implemented as follows. First, artificial graphite (purity > 99%, Knano Graphene Technology, Xiamen, China) and MWCNTs (purity > 99%, Jiangsu Cnano Technology, Zhenjiang, China) were used as feed materials for the delamination experiment. All materials were used as supplied without further purification. Deionized water was used for the preparation of all graphite suspensions and graphite/CNTs (5 wt.%). All delamination experiments were conducted in a low-temperature ultra-high-pressure continuous flow cell disrupter (JNBIO, JN 10C, Guangzhou, China). In this device, a graphite and graphite/CNT suspension was prepared through high pressure forcing the sample through a small orifice at high speed. Because of the shearing, hole, and impact, the graphite in the sample was crushed; the substances in the sample were dispersed and emulsified. FLG would be kept in suspension after liquid exfoliation at 14 °C–16 °C in a circulating water bath. Deionized water was used as a solvent. Graphene production was performed in two batch runs at 800 bar and two batch runs at 1200 bar. The as-synthesized FLG and the FLG/CNT suspension were dried through oven drying or freeze-drying processes. The condition for oven drying was 80 °C and the freeze-drying conditions were −55 °C under a pressure of 20 Pa. The final samples were oven-dried (FLG, GNT10) or freeze-dried (FFLG, FGNT10).

### 2.3. Preparation of EMI Shielding Testing Plates

FLG or FLG/CNT powder was mixed with paraffin by 15 wt.% at 180 °C for 2 h. Subsequently, the mixture was poured into a mold (3.5 cm × 3.5 cm), and pressure of 17,000 pounds (1.5 × 10^6^ N/m^2^) was applied for 30 min. Finally, the bulk material was removed for electromagnetic-wave-shielding measurements.

### 2.4. Characterizations

The surface morphology and crystal structure of the materials were characterized using field emission scanning electron microscopy (SEM, JEOL/JSM-7600F, Tokyo, Japan), high-resolution scanning transmission electron microscopy (TEM, JEOL-JEM2100, Tokyo, Japan), and X-ray diffraction (XRD, Bruker D8, Billerica, MA, USA). Raman characterization was performed using a LabRAM ARAMIS Raman confocal microscope (HORIBA JobinYvon, Edison, NJ, USA) with 532-nm laser excitation. The lateral dimensions of FLG and CNTs were evaluated. The specific surface area was determined using the Brunauer–Emmett–Teller equation (Micromeritics Tristar 3000, Norcross, GA, USA). A four-point probe tester (KeithLink LRS4-T) was used for electrical performance analysis to measure the sheet resistance of FLG and CNTs and electrical conductivity of the nanocomposite. The sheet resistance and electronic conductivity were measured 10 times. Atomic force microscope (AFM) images were captured by a Bruker Dimension Icon. The samples for AFM were prepared by dropping the dispersion directly onto freshly cleaved mica wafer with an injector. The microwave scattering parameters (S11 and S21) of FLG and CNTs were measured using a vector network analyzer (Keysight E5071C) in the X-band frequency range of 8.2–12.4 GHz. The nanocomposites were sandwiched between the holder (22.86 mm × 10.16 mm) within the measurement device. The SE of total EMI shielding (SE_T_), microwave absorption (SE_A_), and microwave reflection (SE_R_) were calculated using parameters S_11_ and S_21_ as follows: R = |S_11_|^2^, T = |S_21_|^2^ and A = 1 − R − T. The effective absorbance (A_eff_) can be expressed as A_eff_ = (1 − R − T)/(1 − R). The total EMI SE (SE_T_) is the sum of the reflection from the material surface (SE_R_), absorption of electromagnetic energy (SE_A_), and multiple internal reflections (SE_M_) of electromagnetic radiation. SE_T_ can thus be described as follows: SE_T_ = SE_R_ + SE_A_ + SE_M_

When SE_T_ > 15 dB, SE_M_ can be ignored. Therefore, SE_T_ is usually defined as follows:SE_T_ ≈ SE_R_ + SE_A_SE_R_ and SE_A_ can be calculated as follows:SE_R_ = −log10(1 − R)SE_A_ = −log10(1 − A_eff_) = −log10[T/(1 − R)]

## 3. Results and Discussion

Figure 1a,b shows the surface morphology of graphite and FLG, respectively. The lateral size of the graphite was 2–5 μm. The morphology of the graphite was clearly thick and irregular. Liquid exfoliation was performed to obtain FLG, which had a smaller lateral size and thickness of <10 nm. This indicated that the initial graphite could be effectively exfoliated into thin sheets through a jet cavitation device. A thickness line scan and an atomic force microscopy image of FLG are displayed in Figure 1c,d, respectively. The thickness of FLG was approximately 3 nm (~10 layers) [51,52].

Figure 2a,b presents the SEM images of the FLG/CNT composites obtained through conventional drying, namely FLG and GNT10, respectively. Under optimized operating conditions, single-layer, double-layer, and multilayer chemically converted graphene may be differentiated based on their opacity in SEM images. CNTs were used to intercalate the graphene, and FLG combined with CNTs was coded as GNT10. After the CNTs had been added, it was observed that the CNTs were uniformly distributed on the surface of FLG. Figure 2c displays a TEM image of FLG, which exhibited the few-layer characteristics of graphene sheets. The TEM of GNT10 (Figure 2d) revealed that CNTs were intercalated between the graphene layers. This was attributed to the mechanical liquid-phase exfoliation method enabling the CNTs to be intercalated while expanding the few layers of graphene [53,54]. Thus, the CNTs were evenly distributed throughout FLG.

Figure 3a–d displays the SEM and TEM images of the FLG/CNT composites obtained through freeze-drying, namely FFLG and FGNT10, respectively. Conventional oven drying caused restacking of FLG (Figure 2a), resulting in a poor SE. After freeze-drying, however, surface morphology was smooth, with larger interlayer spacing, as shown in Figure 3a,c. The TEM image also indicates that the CNTs were distributed throughout FFLG, and that the layers were fewer (Figure 3b,d). The uniform dispersion of CNTs on FFLG, shown in Figure 3b,d, was due to the solvent being evaporated directly during the freeze-drying process, whereas only the solvent on the surface of FFLG was evaporated during conventional oven drying. Thus, the graphene layers were restacked, and the SE was lower.

Figure 4 displays a representative Raman spectrum of FLG (excited at 532 nm). The spectrum included a strong graphene band at 1594 cm^−1^, corresponding to graphite. After exfoliation, the spectrum of FLG had a strong graphene band at 1578 cm^−1^. The I_D_/I_G_ ratio for FLG slightly increased from 0.131 to 0.133 after the jet cavitation process. The position of the peak corresponding to graphene was shifted nearly 20 cm^−1^ from 1594 to 1578 cm^−1^. Graphite was thus successfully exfoliated into FLG [38]. Conversely, the graphene bands in the spectra of FFLG and FGNT10 were shifted from 1557 and 1566 cm^−1^, respectively. The I_D_/I_G_ ratio for FGNT10 was 0.19. This was because a Raman deviation in the low-angle direction represents a decrease in the number of graphene layers.

The adsorption–desorption isotherms and corresponding pore size distributions of a series of FLG-based materials are presented in Figure 5. According to Figure 5a,c, the specific surface area of FFLG (30.6 m^2^/g) was higher than that of FLG (29.7 m^2^/g). As shown in Figure 5b, the distribution contained two peaks at approximately 4 and 50 nm. Figure 5d shows a narrow peak at approximately 3 nm and a broad peak at approximately 100 nm. The volume of the freeze-dried macropores was lower for the freeze-dried samples than for the oven-dried samples. The specific surface areas of GNT10 and FGNT10, shown in Figure 5e,g, were measured to be 49.8 and 51.0 m^2^/g, respectively. The greater specific surface area may be due to the introduction of CNTs to prevent restacking of FLG. Figure 5f,h reveals no difference in the pore size distributions of GNT10 and FGNT10. To determine the percentages of micropores, mesopores, and macropores for these samples, we obtained the three corresponding pore distributions by using the BJH method, which are displayed in Figure 5i. Based on the pore size distributions of these FLG-based materials, these materials mainly contained macropores and mesopores. The specific surface area of FLG increased from 29.7 to 30.6 m^2^/g (an increase of 1.54%) through freeze-drying and could be markedly enhanced to 49.8 m^2^/g (an increase of 67.54%). The FGNT10 sample had 51% macropores, 49% mesopores, and 0% micropores. The composition of the material (FLG or FLG/CNTs) and the drying process (oven drying or freeze-drying) thus have critical effects on both the specific surface area and pore size distribution.

The EMI SE of FLG is shown in Figure 6a. FFLG and FGNT10 exhibited favorable absorption. Comparing GNT10 with FLG, the CNT-intercalated material exhibited higher absorption efficiency—22 versus 15 dB. Figure 6b shows the shielding benefit provided by reflection. The material we used, graphene, did not contribute considerably to the reflection benefit. Figure 6c illustrates the total SE, which is a combination of the results shown in Figure 6a,b. Figure 6d displays a histogram revealing the contributions made to EMI shielding in terms of absorption versus reflection for the graphite, FLG, FFLG, GNT10, and FGNT10 samples. FGNT10 provided an SE of nearly 36 dB. Moreover, the addition of CNTs to FLG to obtain GNT10 increased the SE from 15 to 22 dB. For few-layer graphene, most of the contribution to EMI shielding was from absorption. Graphite became restacked during oven drying; thus, the SE decreased from 17 to 15 dB. The EMI SE was improved to as high as 35 dB by the freeze-drying process.

A schema of electromagnetic wave shielding using GNT10 is presented in Figure 7a. Oven drying causes restacking of graphene layers during the drying process. Figure 7b displays a schema of the shielding achieved using FGNT10. EMI is attenuated through three major mechanisms: reflection, absorption, and multiple reflections. In cases where the shielding effect through absorption (i.e., absorption loss) is higher than 10 dB, most of the re-reflected waves are absorbed within the shield.

Figure 8 shows the four-point probe measurement results for FLG, FFLG, GNT10, and FGNT10. CNT addition and freeze-drying improved the sheet resistance of material flakes. Furthermore, the addition of CNTs increased interlayer spacing and sheet resistance. Freeze drying did not cause the restacking of graphene layers; thus, the sheet resistance of the FGNT10 sample was 1.25 × 10^−1^ Ω/■.

## 4. Conclusions

In summary, we proposed a liquid exfoliation method for preparing high-quality FLG. By introducing CNTs into FLG, the phenomenon of graphene restacking caused by oven drying can be avoided. The freeze-drying process increases the material’s porosity and solves the problem of graphene restacking. The EMI shielding of FGNT10 was as high as 36 dB. By combining FLG, CNTs, and the freeze-drying approach, we obtained a sample, GNT10, with potential for EMI shielding in the range of 8–12 GHz.

## Data Availability

The data presented in this study are available on request from the corresponding author.

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
