# Peer review of "Facile and Green Process to Synthesize a Three-Dimensional Network Few-Layer Graphene/Carbon Nanotube Composite for Electromagnetic Interference Shielding"

_polymers, 2022, doi:10.3390/polym14091892_

Round 1

Reviewer 1 Report

The manuscript titled “Facile and green process to synthesize a three-dimensional network few-layer graphene/carbon nanotube composite for electromagnetic interference shielding”, reports the fabrication, through an environmentally friendly liquid exfoliation approach and subsequent freeze-drying process, of a three-dimensional carbon-based network by using few-layer graphene and carbon nanotubes for electromagnetic interference shielding applications. These nanostructures were characterized with X-ray diffraction, scanning electron microscopy, energy-dispersive X-ray spectroscopy, and Raman, EMI shielding tests. Furthermore, the ability of these systems to shield electromagnetic interference and the influence of two drying processes (oven drying and freeze-drying) on this property was evaluated. The publication of this manuscript can be recommended in Polymers if the major comments could be addressed properly. Here are some suggestions for the revised version:

  • Figure 1 should be renamed “Scheme 1” as it shows the composite materials preparation scheme. Therefore, subsequently, renumber all the figures.
  • In figure 2 (d), the authors show AFM measurements but do not describe this technique in the “characterization” paragraph. Therefore, they should include the description of that technique, the instrument and the operating conditions used.
  • The AFM measurements also allow to obtain other useful information (see and insert the following articles Nanomaterials 2020, 10, 2549; Chem. Eur. J. 2017, 23, 14937 – 14943.), the authors should also show the surface roughness values and the peak-to-peak of exfoliated systems. In addition, useful information on thickness visually is obtained from the 3D representation of the AFM measurements. Authors should include this additional information to improve and complete this article.
  • In the abstract, the authors wrote that they have characterized the materials by energy dispersion X-ray spectroscopy (EDX), but this technique is not described in the “characterization” paragraph and furthermore no measurements are shown in the article. The use of this technique is important for showing the chemical map of materials. Authors should enter the surface chemical analysis of composites to determine whether they remain impurities on the surface. In addition, I recommend that you also insert TEM as the technique used in the abstract.
  • In the paragraph “characterization” the authors should report the operating conditions for the TEM measurements, how the samples were prepared with the analyzes and the type of grids used, while in the results and discussion add some further details (see and add the following references J. Phys. Chem. C 2008, 112, 22, 8192-8195; ACS Appl. Nano Mater. 2020, 3, 8182-8191.)
  • I advise authors to add in the introduction some information on the properties and important applications of carbon-based materials. For example, I suggest adding the following references Chem. Rev. 2020, 120, 9363−9419; Molecules, 2020, 25, 5731; Polymers 2022, 14, 542; Nanomaterials 2020, 10, 2549.

Author Response

The manuscript titled “Facile and green process to synthesize a three-dimensional network few-layer graphene/carbon nanotube composite for electromagnetic interference shielding”, reports the fabrication, through an environmentally friendly liquid exfoliation approach and subsequent freeze-drying process, of a three-dimensional carbon-based network by using few-layer graphene and carbon nanotubes for electromagnetic interference shielding applications. These nanostructures were characterized with X-ray diffraction, scanning electron microscopy, energy-dispersive X-ray spectroscopy, and Raman, EMI shielding tests. Furthermore, the ability of these systems to shield electromagnetic interference and the influence of two drying processes (oven drying and freeze-drying) on this property was evaluated. The publication of this manuscript can be recommended in Polymers if the major comments could be addressed properly. Here are some suggestions for the revised version:

  • Figure 1 should be renamed “Scheme 1” as it shows the composite materials preparation scheme. Therefore, subsequently, renumber all the figures.

Response:

Thank you for reviewer’s suggestion. We have revised it and renumbered all the figures in the revised manuscript.

  • In figure 2 (d), the authors show AFM measurements but do not describe this technique in the “characterization” paragraph. Therefore, they should include the description of that technique, the instrument and the operating conditions used.

Response:

Thank you for reviewer’s comment. We have added the description of the technique. The instrument and the operating conditions used in the revised manuscript.

“Atomic force microscope (AFM) images were captured by a Bruker Dimension Icon. The samples for AFM were prepared by dropping the dispersion directly onto freshly cleaved mica wafer with an injector.”

  • The AFM measurements also allow to obtain other useful information (see and insert the following articles Nanomaterials 2020, 10, 2549; Chem. Eur. J. 2017, 23, 14937 – 14943.), the authors should also show the surface roughness values and the peak-to-peak of exfoliated systems. In addition, useful information on thickness visually is obtained from the 3D representation of the AFM measurements. Authors should include this additional information to improve and complete this article.

Response:

Thank you for reviewer’s valuable comments. The good references have been added in the revised manuscript. Graphene is a two-dimensional material. We try to obtain thickness information of our synthesized graphene. Normally, some researcher use 3D representation of the AFM measurements to show the roughness of the sample. Thus, line scan is enough in this study to demonstrate the thickness information of as-synthesized few layer graphene. We hope reviewer could satisfy with our response.

[51] Salvatore V. Giofrè, Matteo Tiecco, Consuelo Celesti, Salvatore Patanè, Claudia Triolo, Antonino Gulino, Luca Spitaleri, Silvia Scalese, Mario Scuderi and Daniela Iannazzo, “Eco-Friendly 1,3-Dipolar Cycloaddition Reactions on Graphene Quantum Dots in Natural Deep Eutectic Solvent,” Nanomaterials 2020, 10, 2549.

[52] Annalinda Contino, Giuseppe Maccarrone, Maria E. Fragalà, Luca Spitaleri, Antonino Gulino, “Conjugated Gold–Porphyrin Monolayers Assembled on Inorganic Surfaces,” Chem. Eur. J. 2017, 23, 14937 – 14943.

  • In the abstract, the authors wrote that they have characterized the materials by energy dispersion X-ray spectroscopy (EDX), but this technique is not described in the “characterization” paragraph and furthermore no measurements are shown in the article. The use of this technique is important for showing the chemical map of materials. Authors should enter the surface chemical analysis of composites to determine whether they remain impurities on the surface. In addition, I recommend that you also insert TEM as the technique used in the abstract.

Response:

We are very sorry for the typo mistake. We have removed “EDX” in the text. Thank you for reviewer’s kindly reminding. Because for few-layer graphene (FLG), there is only carbon (or trace oxygen) in the material. EDX may not be an important tool to identify the properties of FLG in this study. Sure, we have inserted TEM information in the revised abstract.

  • In the paragraph “characterization” the authors should report the operating conditions for the TEM measurements, how the samples were prepared with the analyzes and the type of grids used, while in the results and discussion add some further details (see and add the following references J. Phys. Chem. C 2008, 112, 22, 8192-8195; ACS Appl. Nano Mater. 2020, 3, 8182-8191.)

Response:

Thank you for reviewer’s information. The details of TEM sample preparation and the corresponding important references have been added in the revised manuscript.

[53] Wang G.; Yang J.; Park J.; Gou X.; Wang B., Liu H.; Yao J. Facile synthesis and characterization of graphene nanosheets. J. Phys. Chem. C 2008, 112, 22, 8192-8195.

[54] Tuccitto N.; Riela L.; Zammataro A.; Spitaleri L.; Li-Destri G.; Sfuncia G.; Nicotra G.; Pappalardo A.; Capizzi G.; Sfrazzetto T. G. Functionalized Carbon Nanoparticle-Based Sensors for Chemical Warfare Agents. ACS Appl. Nano Mater. 2020, 3, 8182-8191.

  • I advise authors to add in the introduction some information on the properties and important applications of carbon-based materials. For example, I suggest adding the following references Chem. Rev. 2020, 120, 9363−9419; Molecules, 2020, 25, 5731; Polymers 2022, 14, 542; Nanomaterials 2020, 10, 2549.

Response:

Thank you for reviewer providing a series of carbon-based references. Theses references have been added in the revised manuscript.

[26] Wang H.; Shao Y.; Mei S.; Lu Y.; Zhang M.; Sun J. -K.; Matyjaszewski K.; Antonietti M.; Yuan J. Polymer-derived heteroatom-doped porous carbon materials. Chem. Rev. 2020, 120, 9363−9419.

[27] Tuccitto N.; Spitaleri L.; Li-Destri G., Pappalardo A.; Gulino A.; Sfrazzetto T. G. Supramolecular sensing of a chemical warfare agents simulant by functionalized carbon nanoparticles. Molecules, 2020, 25, 5731.

[28] Jang D.; Park J. -E., Kim Y. –K. Evaluation of (CNT@CIP)-embedded magneto-resistive sensor based on carbon nanotube and carbonyl iron powder polymer composites. Polymers, 2022, 14, 542.

[51] Giofrè S. V.; Tiecco M.; Celesti C.; Patanè S.; Triolo C., Gulino A.; Spitaleri L.; Scalese S.; Scuderi M.; Iannazzo D. Eco-friendly 1,3-dipolar cycloaddition reactions on graphene quantum dots in natural deep eutectic solvent. Nanomaterials 2020, 10, 2549.

Reviewer 2 Report

  1. Abstract need to be improved to present the significant outcome and innovation points of this study.
  2. In your paper, please highlight the following:
    why this work has been done,
    what is new about it,
    highlight the novelty (has such work been done before, if so, why you are doing this work)
  3. Introduction- What is the role of the carbon-based composite materials in EMI shielding features. Need to be elaborated.
  4. There are many errors in the paper, so the Authors are encouraged to review the form and the English of the manuscript.
  5. or Figure X must be uniformly cited in the manuscript.
  6. Sorption measurements- No significant changes observed? Author needs to elaborate on this?
  7. When presenting the results, compare EMI performance with the latest literature. After showing the results, it is recommended to provide a short conclusion to the obtained results
  8. Many space errors/punctuation errors must be solved. The abbreviations should be checked in the manuscript and made clear.
  9. Conclusion section- must focus on future directions of these composite materials? General statements need to be removed. Authors are suggested to be more specific in their conclusions.

Author Response

  1. Abstract need to be improved to present the significant outcome and innovation points of this study.

Response:

Thank you for reviewer’s suggestion. We have revised abstract to present the significant outcome and innovation of this study.

  1. In your paper, please highlight the following:
    why this work has been done,
    what is new about it,
    highlight the novelty (has such work been done before, if so, why you are doing this work)

Response:

Thank you for reviewer’s comments. We have highlighted these items in the revised manuscript.

  1. Introduction- What is the role of the carbon-based composite materials in EMI shielding features. Need to be elaborated.

Response:

We have revise the information based on reviewer’s comments.

  1. There are many errors in the paper, so the Authors are encouraged to review the form and the English of the manuscript.

Response:

The English of the manuscript has been revised and edited. Please see the certificate.

  1. or Figure X must be uniformly cited in the manuscript.

Response:

We are sorry for the mistakes. Figure X have been uniformly cited in the revised manuscript.

  1. Sorption measurements- No significant changes observed? Author needs to elaborate on this?

Response:

Thank you for reviewer’s suggestion. We have elaborated it in the revised manuscript.

  1. When presenting the results, compare EMI performance with the latest literature. After showing the results, it is recommended to provide a short conclusion to the obtained results

Response:

Thank you for reviewer’s suggestion. We have provided a short conclusion in the revised manuscript.

  1. Many space errors/punctuation errors must be solved. The abbreviations should be checked in the manuscript and made clear.

Response:

We are sorry for the mistakes. We have corrected these errors and mistakes in the revised manuscript. The corresponding abbreviations have also been checked in the manuscript.

  1. Conclusion section- must focus on future directions of these composite materials? General statements need to be removed. Authors are suggested to be more specific in their conclusions.

Response:

Thank you for reviewer’s suggestion. We have revised the conclusion section in the revised manuscript.

Reviewer 3 Report

This study is devoted to preparing the composites based on few-layer graphene and carbon nanotubes for electromagnetic interference shielding. The authors have utilized SEM, TEM, XRD, Raman microscopy, N2 adsorption at 77 K, and other techniques to characterize the prepared materials. As a result, they have obtained the composite with the electromagnetic interference shielding effect up to 40 dB. The main advantage of the provided work is its practical significance because the obtained material can be used to protect humans against the dangerous electromagnetic waves upon using the gadgets. However, in this paper, the structure of the resulted composite materials and the possible mechanisms of neutralizing the electromagnetic waves by these materials are also discussed in depth. Generally, the manuscript is well-written and thoroughly prepared. Therefore, I recommend this article be accepted for publishing in Polymers after the authors perform the following minor revisions:

  1. Lines 95-96: Please, check the style of this sentence.
  2. Line 110: Please, check the spelling of the value and corresponding unit.
  3. Lines 112-137: Please, provide more detailed procedures for the measurements.
  4. Line 119: Please, write that the specific surface area was measured utilizing low-temperature nitrogen adsorption and determined using the BET equation.
  5. Lines 136, 137: Please, check indexes in the formulae.
  6. Line 143: Probably, a dot is lost here.
  7. Line 186: The authors should provide a more detailed discussion of the measured isotherms, including their type.
  8. Line 186: Which method was used to calculate these distributions?
  9. Lines 188, 193, 200: All numbers after a dot in the values of the specific surface area do not have any sense because the error of estimating the specific surface area values is not less than 1 m2/g.
  10. Line 189: The word “isotherm” should be replaced with “curve” or “distribution”.
  11. Line 189: Were the studied nanotubes open-tubular or closed-tubular? The measured values of the specific surface areas are low and witness the absence of the micropores in the prepared materials. Sometimes, the peaks related to micropores in the pore sizes distributions can be artifacts.
  12. Line 194-195, 200: These values of the specific surface area that the authors compare are rather the same (please, see note 9).
  13. Please, compare lines 199 and 202. Do you talk about micropores or macropores?
  14. Please, check the title of Figure 6. Expressly, it is wrong.
  15. Line 222: Please, decipher the acronyms.
  16. Line 236: Please, check the value’s unit.
  17. Lines 239-245: Please, do not use acronyms in this section to improve its readability.

Author Response

  1. Lines 95-96: Please, check the style of this sentence.

Response:

We have checked and corrected the style of the sentence.

  1. Line 110: Please, check the spelling of the value and corresponding unit.

Response:

We have checked the spelling of the value and corresponding unit.

  1. Lines 112-137: Please, provide more detailed procedures for the measurements.

Response:

Thank you for reviewer’s suggestion. We have provided more detailed procedures for these measurements.

  1. Line 119: Please, write that the specific surface area was measured utilizing low-temperature nitrogen adsorption and determined using the BET equation.

Response:

In line 119, the geometry of the holder was 22.86 mm × 10.16 mm. We put testing sample into the holder. So, it may not need to measure the specific surface area of the holder. We hope reviewer could satisfy with our response.

  1. Lines 136, 137: Please, check indexes in the formulae.

Response:

Thank you for reviewer’s suggestion. The indexes in the formulae have been checked and corrected in the revised manuscript.

  1. Line 143: Probably, a dot is lost here.

Response:

Thank you for reviewer’s comment. Yes. We have added a dot in the text.

  1. Line 186: The authors should provide a more detailed discussion of the measured isotherms, including their type.

Response:

Thank you for reviewer’s suggestion. We have provided more detailed discussion of the measured isotherms, including the type in the revised manuscript.

  1. Line 186: Which method was used to calculate these distributions?

Response:

The pore size distribution was obtained by using the BJH method. We will mention it in the revised manuscript. Thank you for reviewer’s comments.

  1. Lines 188, 193, 200: All numbers after a dot in the values of the specific surface area do not have any sense because the error of estimating the specific surface area values is not less than 1 m2/g.

Response:

Thank you for reviewer’s comments. The numbers of these specific area values have been corrected in the revised manuscript.

  1. Line 189: The word “isotherm” should be replaced with “curve” or “distribution”.

Response:

Yes. It should be distribution. The mistake has been corrected in the revised manuscript.

  1. Line 189: Were the studied nanotubes open-tubular or closed-tubular? The measured values of the specific surface areas are low and witness the absence of the micropores in the prepared materials. Sometimes, the peaks related to micropores in the pore sizes distributions can be artifacts.

Response:

Thank you for reviewer’s comment. We agree with reviewer’s points of view. Sometimes, the peaks related to micropores in the pore sizes distributions can be artifacts. We cannot identify its open-tubular or closed-tubular. For graphene, non-porous material, there is little “pore” in the structure. Thus, we don’t need to mention the type of “open-tubular” or “closed-tubular” in the text.  

  1. Line 194-195, 200: These values of the specific surface area that the authors compare are rather the same (please, see note 9).

Response:

Yes. The specific surface of FLG and FFLG was similar. It means that by using oven drying process and freeze drying process, no rather enhancement was observed. But, if we introduce CNTs into the composite, the specific surface area was dramatically increased. The effects of CNTs was obvious compared to drying condition effects.

  1. Please, compare lines 199 and 202. Do you talk about micropores or macropores?

Response:

Thank you for reviewer’s reminding. Yes. It should be macropores. The typo error has been corrected in the revised manuscript.

  1. Please, check the title of Figure 6. Expressly, it is wrong.

Response:

We are sorry for the mistakes. The correct caption of Fig. 6 (Revised Fig. 5) is

Figure 6. Adsorption–desorption isotherms (a, c, e, g), corresponding pore size distributions (b, d, f, h) and (i)micropores/mesopores/macropores distribution of graphite, FLG, FFLG, GNT10, and FGNT10.

Which is corrected in the revised manuscript.

  1. Line 222: Please, decipher the acronyms.

Response:

These acronyms have been deciphered in line 130-131.

  1. Line 236: Please, check the value’s unit.

Response:

The unit of the value (W/n) is correct.

  1. Lines 239-245: Please, do not use acronyms in this section to improve its readability.

Response:

Thank you for reviewer’s suggestion. We revised the conclusion part to improve its readability.

Round 2

Reviewer 1 Report

Authors followed and responded satisfactorily to all my suggestions. The manuscript is now improved and can be accepted for publication.